# Comparative Effectiveness of Pediatric Integrative Medicine: A Pragmatic Cluster-Controlled Trial

**DOI:** 10.3390/children8040311

**Published:** 2021-04-20

**Authors:** Sunita Vohra, Salima Punja, Hsing Jou, Michael Schlegelmilch, Beverly Wilson, Maria Spavor, Paul Grundy, Andrew S. Mackie, Jennifer Conway, Dawn Hartfield

**Affiliations:** 1Faculty of Medicine and Dentistry, 3-548 Edmonton Clinic Health Academy (ECHA), 11405-87 Avenue, Edmonton, AB T6G 1C9, Canada; 2Faculty of Medicine and Dentistry, University of Alberta, Edmonton, AB T6G 2R3, Canada; punja@ualberta.ca (S.P.); hjou@ualberta.ca (H.J.); mschlege@ualberta.ca (M.S.); bev.wilson@albertahealthservices.ca (B.W.); mspavor@ualberta.ca (M.S.); pgrundy@ualberta.ca (P.G.); mackie1@ualberta.ca (A.S.M.); conway@ualberta.ca (J.C.); dawn.hartfield@cpsa.ab.ca (D.H.)

**Keywords:** pediatrics, integrative medicine, complementary therapies

## Abstract

Symptoms of pain, nausea/vomiting, and anxiety (PNVA) are highly prevalent in pediatric inpatients. Poorly managed symptoms can lead to decreased compliance with care, and prolonged recovery times. Pharmacotherapy used to manage PNVA symptoms is of variable effectiveness and carries safety risks. Complementary therapies to manage these symptoms are gaining popularity due to their perceived benefits and low risk of harm. Pediatric integrative medicine (PIM) is the combination of complementary therapies with conventional medicine in pediatric populations. A two-arm, cluster-controlled, pragmatic clinical trial was carried out to compare the effectiveness of a PIM service in conjunction with usual care, versus usual care only to treat PNVA symptoms in hospitalized pediatric patients. The primary outcome was the improvement of PNVA symptom severity using a 10-point numerical rating scale. Participant enrollment occurred between January 2013 and January 2016. A total of 872 participants (usual care *n* = 497; PIM *n* = 375) were enrolled. The PIM therapies significantly reduced PNVA symptom severity (*p* < 0.001). This study found that a hospital-based PIM service is both safe and effective for alleviating PNVA symptoms. Future research should carry out this work in other pediatric inpatient divisions, and in other sites to determine the reproducibility of findings.

## 1. Introduction

Symptoms of pain, nausea/vomiting, and anxiety (PNVA) are highly prevalent in pediatric inpatients across North America. Pain is common, under-recognized, and under-treated in pediatric inpatients, and has adverse physiological and psychological effects [1,2]. A survey at a large children’s hospital in Canada reported that 77% of pediatric inpatients experience pain at some time during their hospital stays [3]. Nausea and vomiting are especially common in pediatric populations, particularly oncology: radiation- or chemotherapy-induced nausea/vomiting occurs in over 90% of pediatric oncology patients [4,5]. Anxiety is often associated with both pain and nausea/vomiting, and with hospitalization experiences, both general and procedural. Anxiety is often left untreated in hospitalized children, despite its widespread occurrence [6,7,8]. Poorly managed symptoms can lead to decreased compliance with health care, prolonged recovery times, and increased costs to the health care system [9,10].

Pharmacotherapy used to manage PNVA symptoms is of variable effectiveness and carries safety risks, including serious adverse reactions [9,10,11]. As a result, complementary therapies used to manage these symptoms are gaining popularity due to their perceived benefits and low risk of harm [12,13]. It is estimated that nearly half of children use some form of complementary therapy, with estimates reaching over 70% in children living with serious, chronic, or recurrent conditions [14,15,16]. Specifically, for PNVA symptoms, various complementary therapies have demonstrated safety and efficacy, including acupuncture, massage therapy, and Reiki.

In this study, we focus on three complementary therapy modalities: acupuncture, massage therapy, and Reiki. Acupuncture is a key component of traditional Chinese medicine in which thin needles are applied at specific points throughout the body [17]. Evidence has demonstrated acupuncture to be effective for symptoms of pain and nausea/vomiting [18,19,20]. Massage therapy is defined as a therapeutic manipulation using the hands or mechanical device, which includes numerous techniques that are often used in sequence, such as stroking, kneading, and percussion [21]. Evidence has shown that massage therapy has positive effects on symptoms of pain, nausea/vomiting, and anxiety [22,23]. Finally, Reiki is a therapy that claims to provide healing energy to recharge and rebalance the human energy fields, creating optimal conditions needed by the body’s natural healing system [24]. Reiki has been shown to be effective for symptoms of pain and anxiety [24,25,26]. Despite the evidence that exists, pediatric data on integrative approaches to health care delivery are needed.

Integrative medicine is the combination of evidence-based complementary therapies with conventional medicine in a coordinated, patient-focused way, with the goal of achieving optimal health and healing [27]. Pediatric integrative medicine (PIM) applies this approach to pediatric populations [28]. While many pediatric hospitals across North America have begun to integrate complementary therapies into conventional care [29], there is a lack of conclusive evidence around the safety and effectiveness of this approach.

This study compares the effectiveness of a PIM service that offers three complementary therapy modalities (i.e., acupuncture, massage therapy, Reiki) in conjunction with usual care versus usual care alone to treat PNVA symptoms in hospitalized pediatric patients. To our knowledge, this is the first comparative effectiveness trial to assess the impact of PIM for hospitalized children.

## 2. Materials and Methods

The institutional review board of the University of Alberta approved the protocol (Pro00010904) and the trial is registered on clinicaltrials.gov (identifier NCT02028832).

This was a mixed-methods study including quantitative, qualitative, and health economics analyses. This paper presents the quantitative results.

Study methods will be briefly discussed here; however, a full description of the study methods has been previously published [30].

### 2.1. Design

A two-arm, cluster controlled, pragmatic trial. Each participating hospital division received PIM services or not for 6 months each in an ABA design, where A is usual care and B is usual care in conjunction with PIM.

### 2.2. Study Participants

This study was conducted at the Stollery Children’s Hospital in Edmonton, Alberta, Canada, in three inpatient divisions: pediatric oncology, general pediatrics, and pediatric cardiology. The Stollery Children’s Hospital is a 218-bed, full-service pediatric hospital and centre for complex pediatric care and research. The Stollery has among the highest inpatient volumes of any children’s hospital in Canada, and serves a geographical area of over 500,000 km [31].

Participants were eligible if they were admitted into a participating division, <17 years of age upon admission, and had a length of stay between 2 and 30 days. Participants who had a hospital length of stay for >30 days were excluded from the analysis.

### 2.3. Treatment Arms

#### 2.3.1. Control: Usual Care

Usual care included conventional care including pharmacotherapy and follow-up as per usual division-specific standards.

#### 2.3.2. Intervention: PIM Service

The PIM service has been described previously [29]. Briefly, this arm received usual care in conjunction with PIM therapies including acupuncture, Reiki, and massage therapy. If a child was symptomatic for any of the study’s target symptoms (PNVA), they were eligible to be seen the same day by PIM staff. Based on the child’s needs and preferences, and in light of best evidence therapeutic indications, recommendations were made for them to receive one or more of the PIM therapies by credentialed providers. Participants were followed until their symptoms were resolved and re-consultation with the PIM service for a new or recurring target symptom was offered on an as-needed basis.

### 2.4. Blinding and Bias

Due to the nature of the PIM therapies, blinding was not possible. Moreover, since this study assesses subjective symptoms, extra methodological precautions were put in place to minimize potential bias, including: (i) careful documentation of participant characteristics in order to control for differences between arms including previous PIM therapy use, as well as beliefs/expectations regarding PIM therapies; (ii) use of an active control arm; (iii) utilizing an ABA design that allowed us to ensure that no changes in usual care resulted from having augmented symptom awareness during the PIM period; and (iv) carrying out blinded analysis.

### 2.5. Outcomes and Measures

The primary outcome was change in PNVA symptom scores in the PIM group measured using a 10-point numerical rating scale (NRS) administered immediately prior to and immediately after each PIM therapy intervention.

Secondary outcomes included comparative parent satisfaction upon discharge using the hospital’s standardized satisfaction survey between the two study arms, and difference in hospital length of stay between the two study arms. Adverse events were collected from patients and parents via daily assessments by the research nurse [32].

### 2.6. Sample Size

This study is the first of its kind, and as a result, data on minimally important differences in symptom severity are sparse. We conducted a pragmatic trial, enrolling all eligible consenting participants during the study period. No sample size calculation was possible due to a lack of data about effect size; we aimed to enroll 50 children in each arm (usual care and PIM) of pediatric oncology and cardiology, and 100 in each arm of general pediatric inpatients.

### 2.7. Analysis

Statistical analyses were performed according to intention-to-treat principle. The data from the three participating divisions were combined for the analysis. A *p*-value of <0.05 was considered statistically significant for all comparisons. All analyses were descriptive and exploratory; thus, the statistical significance level for these analyses was not adjusted for multiple testing. Patients’ baseline characteristics, age, sex, verbal status, ancestral background, and history of complementary therapy use were tabulated for the two study arms. Continuous variables were presented as means with standard deviations and categorical variables as counts with percentages. For each symptom of pain, nausea, vomiting, and anxiety, the percentage of days with the symptom and the mean symptom score during admission was computed for each patient. Then, the mean percentage of days with a symptom and the mean average symptom score rating were compared between the treatment groups using Student t-tests. In the PIM group, the number of patient days for the PIM therapy were tabulated by therapy (acupuncture, Reiki, or massage therapy) and by the target symptom. Means of NRS scores averaged over all symptoms before and after receiving each therapy were compared using paired *t*-test. Median parent satisfaction scores on the numerical rating scale were compared between the treatment groups using the Mann–Whitney U test. Mean length of hospital stay was compared between the groups using Students *t*-test. All analyses were performed using IBM SPSS Statistics, Version 25.

## 3. Results

### 3.1. Study Population

Participant enrollment occurred between January 2013 and January 2016. A total of 872 participants (usual care *n* = 497; PIM *n* = 375) across all three divisions were enrolled, of which 861 participants were included in the analysis. See Figure 1 for participant flow through the trial.

Participants were predominantly Caucasian, and had a mean age of 5.0 years in the usual care arm and 5.3 years in the PIM arm. About one half of the enrolled children were female, a third were verbal, and nearly one-third had previously used a complementary therapy (Table 1). The two populations were comparable.

### 3.2. Complementary Therapy Use in the PIM Arm

There were a total of 997 patient days in which a complementary therapy was requested and administered in the PIM arm, massage therapy being the most common (40.4%), followed by Reiki (33.6%) and then acupuncture (26%) (Table 2). The most common reason for a PIM referral was pain (30.8%) followed by general wellbeing (18.2%). Other reasons for PIM referrals, apart from nausea, vomiting, and anxiety, included anorexia, fatigue, constipation, diarrhea, spasms/tension, and non-specific symptoms (Table 3).

### 3.3. Primary Outcome

Symptom severity was reduced across all modalities (*p* < 0.001). Symptoms improved on average by 40.9%, 42.9%, and 48.0% in acupuncture, Reiki, and massage therapy, respectively (Table 4).

### 3.4. Secondary Outcomes

Parental satisfaction was 9/10 and 10/10 in the usual care and PIM arms, respectively (*p* = 0.161). Length of stay did not significantly differ between the usual care and PIM arms (*p* = 0.602) (Table 5).

### 3.5. Safety

According to daily assessments of participants and parents, of the 375 patients in the intervention arm, there was one reported adverse event associated with massage therapy (i.e., transient rash after massage with grape seed oil, which resolved without treatment).

## 4. Discussion

To our knowledge, this is the first comparative effectiveness study of the impact of PIM as adjunctive care of pediatric inpatients. Our findings show that PIM therapies, specifically acupuncture, Reiki, and massage therapy, are very effective at improving PNVA symptoms in pediatric inpatients (40.9% to 48% improvement), with no serious adverse events. While the target symptoms in this study were PNVA, we found that PIM referrals were sought by patients/parents for a variety of other reasons, including general well-being.

These findings are consistent with other PIM studies. In a 10-week study carried out at large children’s hospital to assess a pilot integrative medicine inpatient consult service, 18 patients received the integrative service during their hospitalization [29]. All patients noted a decrease in their pain scores after receiving the integrative therapies, and 100% (*n* = 15) of the interviewed families felt that the service helped both their child’s pain and mood while hospitalized [29]. A pilot study assessing massage therapy in 25 children with cancer found that massage therapy was significantly more effective at reducing anxiety in children compared to quiet time control [33]. A 2009 Cochrane review that included seven randomized controlled trials of children undergoing surgery (*n* = 727) found that acupuncture significantly reduced post-operative nausea and vomiting, and therefore reduced the need for rescue anti-emetics [34]. Moreover, a pre-post mixed methods single group pilot study assessed the impacts of Reiki in children receiving palliative care. The findings showed that Reiki decreased symptoms of pain and anxiety; however, the small sample size (*n* = 16) limited statistical significance [35].

This study has a number of strengths, including a high rate of enrollment and follow-up. Moreover, the pragmatic design of this study enables us to draw conclusions about hospital-based PIM while accommodating the diversity of the three divisions, and allows us to describe the effects of complementary therapies under typical conditions rather than ideal conditions, allowing the results to inform decision-makers on additional options for hospital care [36].

The external generalizability of the results to other children’s hospitals is also high, as we have previously verified that patient demographics, illness, types of treatment, and outcomes of children seen at our hospital is comparable to others in North America [37,38].

There are limitations to this study. Because patients and researchers were not blinded to the interventions, measurement bias may have occurred, thus potentially yielding exaggerated symptom improvements. Reporting bias may also be an issue in this study. Priority of the PIM therapies was given to patients who reported PNVA symptoms, which could account for the significantly higher rates of pain reported in the PIM arm compared to the usual care arm of the study. We attempted to mitigate the potential for symptom inflation by allowing participants to access the PIM service for any symptom of concern, not only PNVA. While it is true that the majority of participants in this study identified as Caucasian (59%), complementary therapies are of great interest to and often used by a variety of ethnicities around the world [39], and as such may have even greater relevance to those within diverse sociocultural backgrounds.

## 5. Conclusions

In this study, we found that a hospital-based PIM service is both safe and effective for alleviating PNVA symptoms. As complementary therapy use continues to gain popularity in pediatric populations, more rigorous studies are required to support the safety and effectiveness of these therapies. The integration of complementary therapies with conventional care for pediatric inpatients offers several opportunities, including the ability to provide truly multidisciplinary care, as well as the ability to counsel patients/parents on the risks and benefits of complementary therapies. Findings from this study support this integration in health care delivery.

Future research should carry out this work in other pediatric inpatient divisions, and in other sites to determine reproducibility of findings. Future studies could also include the assessment of other complementary therapies with favourable risk: benefit profiles such as music, yoga, dance/movement therapy, and mindfulness.

## Figures and Tables

**Figure 1 children-08-00311-f001:**
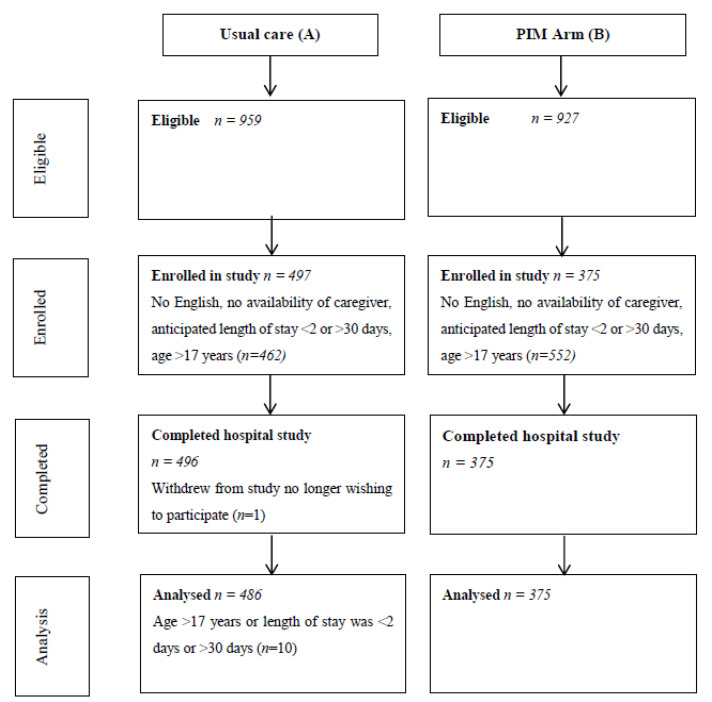
Participant flow.

**Table 1 children-08-00311-t001:** Participant characteristics.

	Usual Care		PIM	
	N	%	N	%
No. of participants	486		375	
Age (year), mean (SD)	5.0 (5.0)		5.3 (5.1)	
Female	229	46.1	198	52.8
Verbal	168	34.6	130	34.7
Ancestral Background				
African	15	3.1	7	1.9
Arabic	3	0.7	7	1.9
Caucasian	286	58.8	230	61.3
Chinese	5	1.0	1	0.3
East Indian, Pakistani, Sri Lankan	19	3.9	12	3.2
Filipino	33	6.8	20	5.3
First Nations/Inuit, Metis	43	8.9	47	12.5
Latin American/Mexican	7	1.4	6	1.6
Identifies with multiple above	67	13.8	37	9.9
Other	8	1.6	8	2.1
Previous CT use (child)	130	26.7	147	39.2

Acronyms: SD, standard deviation; CT, complementary therapy; PIM, Pediatric Integrative Medicine.

**Table 2 children-08-00311-t002:** Complementary therapy use by patient days.

Modality	N (%)
Acupuncture	259 (26.0)
Reiki	335 (33.6)
Massage	403 (40.4)
Total	997 (100)

**Table 3 children-08-00311-t003:** Reasons for PIM referrals.

Primary Target Symptom	
Pain	307 (30.8)
Nausea/Vomiting	128 (12.8)
Anxiety	80 (8.0)
General well-being	181 (18.2)
Other	263 (26.4)
Missing/Not reported	38 (3.8)

Acronyms: Other: Anorexia, lethargy/fatigue, constipation, diarrhea, spasms, non-specific.

**Table 4 children-08-00311-t004:** Pre- and post-treatment NRS symptom scores for acupuncture, Reiki, or massage therapy for any reported symptom in the PIM arm.

Modality	No. of Participants	No. of Observations	Pre-Treatment NRS Score	Post-Treatment NRS Score	*p*-Value	Percent Improvement
Acupuncture	96	332	5.52 ± 2.20	3.26 ± 2.11	<0.001	40.9%
Reiki	112	359	5.52 ± 2.26	3.15 ± 2.10	<0.001	42.9%
Massage therapy	163	426	4.50 ± 1.88	2.34 ± 1.97	<0.001	48.0%

Reported as mean (SD). Acronym: NRS, numerical rating scale.

**Table 5 children-08-00311-t005:** Secondary outcomes: Satisfaction, hospital length of stay.

Indicator	Usual Care	PIM	*p*-Value
Numerical Rating of Satisfaction Score (out of 10)	Median (IQR)		
	9.0 (8.0–10.0)	10.0 (9.0–10.0)	0.161
Length of stay in hospital (days)	Mean (SD)		
	5.9 (9.6)	6.5 (7.8)	0.602

Acronyms: IQR: interquartile range; SD, standard deviation.

## Data Availability

Deidentified individual participant data will not be made available.

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
