# Peer review of "Comparative Effectiveness of Pediatric Integrative Medicine: A Pragmatic Cluster-Controlled Trial"

_children, 2021, doi:10.3390/children8040311_

Round 1
Reviewer 1 Report
Comments by line number:
I will be using "CAM" for complementary therapies
32 - peds onc pt experiencing pain is not representative of all 'pediatric inpatients'; a study showing pain in pediatric inpatients in general would be useful. Peds onc pts likely have higher PNVA, as noted in the following line.
43 - therapies...are gaining popularity
43 - do you have references showing growing use of CAM and potential benefits?
53/54 - 'lack of conclusive evidence around the safety and effectiveness of this approach' - this generalizes CAM. Many CAM therapies are irrefutably safe, while others, as you mention, lack conclusive evidence.
55 - it would be edifying for the reader if you include potential MOAs for these three CAM modalities' effect on PNVA
58 - this is the first
68 - how were the clusters randomized? why were individuals not randomized?
79 - remove "for those"
re: Study participants - was there any exclusion criteria based on how sick the patients were? e.g. peds onc pts may have greater PNVA symptoms and thereby also more likely to respond to CAM
91 - you can use they instead of s/he
148/149 - one-half, one-third
171 - why does table 4 not include usual treatment w/o CAM? what was the efficacy in the no CAM arm?
179 - are there long term safety concerns with these three CAM? any potential interactions between the CAM and the usual care (e.g. drug-drug interactions, which are less likely in this case).
193 - %
Congrats on being able to enroll so many peds pts in this study. Overall; please include more data comparing the CAM group to the usual care group. Table 4: understandably the usual care group w/o CAM probably had lower post-treatment NRS scores but I would like to see that. Then compare the percent improvement to the percent improvement of the +CAM group. Also include an average of these three CAM groups to represent CAM overall. In the introduction or treatment arms, describe the research behind these CAM therapies, e.g. why/how are they effective, their safety. I know they have been described in a previous paper but they are very relevant to this paper.
Author Response
32 - peds onc pt experiencing pain is not representative of all 'pediatric inpatients'; a study showing pain in pediatric inpatients in general would be useful. Peds onc pts likely have higher PNVA, as noted in the following line.
Response: Thanks for catching this. We made a mistake in our reporting of the study we cite. The study we cite actually conducted a cross-sectional survey of all medical and surgical inpatient units at a Children’s hospital, and the 77% who experienced pain represents all units (not simply oncology). We have corrected this on line 33.
43 - therapies...are gaining popularity
Response: This change has been made to line 43.
43 - do you have references showing growing use of CAM and potential benefits?
Response: Additional references have been added.
53/54 - 'lack of conclusive evidence around the safety and effectiveness of this approach' - this generalizes CAM. Many CAM therapies are irrefutably safe, while others, as you mention, lack conclusive evidence.
Response: Thank you. What we mean by this is that there is lack of conclusive evidence around the integration of complementary therapies into conventional care (as opposed to the use of complementary therapies alone).
55 - it would be edifying for the reader if you include potential MOAs for these three CAM modalities' effect on PNVA
Response: We have added a paragraph (lines 55 to 66) that goes into further detail about each of the modalities and evidence to support their effects.
58 - this is the first
Response: We fixed this typo.
68 - how were the clusters randomized? why were individuals not randomized?
We chose an A-B-A design, rather than randomizing clusters, as we wanted to measure symptoms reported in usual care before and after the intervention arm, to see if the trial itself influenced how care was given.
79 - remove "for those"
Response: This has been removed.
re: Study participants - was there any exclusion criteria based on how sick the patients were? e.g. peds onc pts may have greater PNVA symptoms and thereby also more likely to respond to CAM
Response: There were no exclusion criteria based on how sick the patients were. All patients in participating Divisions were seen every morning, to assess symptoms and to determine if any PIM services should be arranged that afternoon. As is evident in the data, the response to PIM was significant across all divisions (not just oncology).
91 - you can use they instead of s/he
Response: This has been edited.
148/149 - one-half, one-third
Response: This has been edited.
171 - why does table 4 not include usual treatment w/o CAM? what was the efficacy in the no CAM arm?
Response: This table reflects the pre- post-treatment scores for the complementary therapies (i.e. what was their symptom score before receiving the treatment, and what was their symptom score after receiving the treatment). Since the usual care arm did not receive complementary therapies, these values would not have been assessed.
179 - are there long term safety concerns with these three CAM? any potential interactions between the CAM and the usual care (e.g. drug-drug interactions, which are less likely in this case).
Response: We chose non-pharmacological approaches to prevent NHP-drug interactions. It is true that our assessment of adverse events was short-term/immediate; we have no knowledge of the participant after they were discharged, but we have no reason to believe there are long-lasting adverse effects associated with the modalities used in the trial.
193 - %
Response: This has been fixed.
Congrats on being able to enroll so many peds pts in this study. Overall; please include more data comparing the CAM group to the usual care group. Table 4: understandably the usual care group w/o CAM probably had lower post-treatment NRS scores but I would like to see that. Then compare the percent improvement to the percent improvement of the +CAM group. Also include an average of these three CAM groups to represent CAM overall. In the introduction or treatment arms, describe the research behind these CAM therapies, e.g. why/how are they effective, their safety. I know they have been described in a previous paper but they are very relevant to this paper.
Response: While daily scores were measured in both arms, pre-post scores were only measured in participants who received PIM interventions (to assess the effect).
We have also added more text in the introduction around the chosen complementary therapies.
Reviewer 2 Report
This is a well organized paper focusing on a very current and important topic. Given the clinical nature of my own work I do not feel qualified to comment on the details of the study design. Listed below are a few suggestions to improve this paper:
- Though I am in agreement with the study's premise that PNVA symptoms continue to be under treated in pediatric patients, the first 10 references are quite old, ranging from 1999 - 2008. Please update providing more recent statistics and procedures about how PNVA symptoms are currently being treated.
- The subjects in this study were "predominantly Caucasian" (line 147). This needs to be referenced as a limitation of the study. Suggestions for future studies should include more diversity and a discussion about how culture and parenting styles may also influence how PNVA symptoms are responded to and treated.
- Line 233 include dance/movement therapy in the list of other complementary therapies to be assessed in future studies.
Author Response
Though I am in agreement with the study's premise that PNVA symptoms continue to be under treated in pediatric patients, the first 10 references are quite old, ranging from 1999 - 2008. Please update providing more recent statistics and procedures about how PNVA symptoms are currently being treated.
Response: We have added some updated references (Stevens 2011; Lerwick 2013)
The subjects in this study were "predominantly Caucasian" (line 147). This needs to be referenced as a limitation of the study. Suggestions for future studies should include more diversity and a discussion about how culture and parenting styles may also influence how PNVA symptoms are responded to and treated.
Response: While it is true that the majority of participants in this study identified as Caucasian (59%), complementary therapies are of great interest to and often used by a variety of ethnicities around the world, and as such may have even greater relevance to those within diverse sociocultural backgrounds.
This text has been added to our discussion (lines 235-238).
Line 233 include dance/movement therapy in the list of other complementary therapies to be assessed in future studies.
Response: This has been added.